# Moonwalk: Inverse-Forward Differentiation

## Abstract

Backpropagation is effective for gradient computation but can requires large memory, limiting scalability. This work explores forward-mode gradient computation as an alternative in invertible networks and, more generally, ones with surjective differentials (submersive networks), showing its potential to reduce the memory footprint without substantial drawbacks. We introduce a novel technique based on a vector-inverse-Jacobian product that accelerates the forward computation of gradients compared to naïve forward-mode methods while retaining their advantages of memory reduction and preserving the fidelity of true gradients. Our method, Moonwalk, has a time complexity linear in the depth of the network, unlike the quadratic time complexity of naïve forward, and empirically reduces computation time by several orders of magnitude without allocating more memory. We further accelerate Moonwalk by combining it with reverse-mode differentiation to achieve time complexity comparable with backpropagation while significantly reducing its memory footprint in some network architectures. Finally, we showcase the robustness of our method across several architecture choices. Moonwalk is the first forward-based method to compute true gradients in submersive networks in computation time comparable to backpropagation and using significantly less memory.

## 1 Introduction

In recent years, the evolution of deep learning models has been significantly influenced by the advent of automatic differentiation (AD) packages (Paszke et al., 2019; Bradbury et al., 2018; Abadi et al., 2015), facilitating expedited model construction and research. The most commonly used differentiation algorithm is backpropagation (Backprop), which effectively addresses the challenge of time-efficient gradient computation, but fails to tackle the issue of memory consumption (Gomez et al., 2017; Chakrabarti & Moseley, 2019). The issue is that, during the construction and reversion of the computation graph, Backprop retains the value of many intermediate computations ("activations") acquired during the forward execution phase, as needed for the backward phase. For large networks, the memory footprint scales with the number of activations (Novikov et al., 2023) which can result in significant memory overhead limiting the ability to scale neural networks.

An alternative to traditional Backprop is forward-mode gradient computation (Forward), a long-established concept in training neural networks (Williams & Zipser, 1989) but one that remains less widely adopted in practice due to its very high computation time. The notion of employing forward differentiation has recently gained attention as a promising strategy for alleviating layer activation memory constraints inherent in Backprop (Silver et al., 2021; Baydin et al., 2022; Fournier et al., 2023). However, a primary drawback of forward gradient computation lies in its requirement to compute full Jacobian matrices for every layer of the computation graph, proving more computationally expensive than the vector-Jacobian products (*vjp*) used in Backprop.

To address this challenge, one potential approach involves projecting true gradients onto a subspace and computing only these projections in forward-mode (Baydin et al., 2022). Some prior works employ projections onto random subspaces, thereby introducing variance in gradient estimation that limits applicability in large networks (Silver et al., 2021). Other works predict the gradient direction based on auxiliary networks or past gradients and use it as a preferred projection subspace, showing promise in reducing variance but to-date falling short of the end-to-end training accuracy of Backprop (Fournier et al., 2023).

In this work, we identify a novel mathematical identity in computing gradients of invertible networks and, more generally, of networks whose layers have differentials that are everywhere surjective, defined in section 3.1 as *submersive networks*. This identity allows significant savings in memory and time when computing true gradients in forward-mode without projection. Our method, Moonwalk, relies on the observation that, once we obtain the gradient of the objective with respect to just the input of the first layer, we can efficiently compute the remaining gradients using a vector-inverse-Jacobian product (*vijp*) operator in forward-mode. To our knowledge, Moonwalk is the first forward-mode differentiation method that can outperform Backprop in both time and memory requirements when computing true gradients in submersive networks.

Moonwalk computes gradients in two phases. In the first phase, it computes the gradient of the objective (the loss) with respect to the first layer's input. In the second phase, Moonwalk uses this input gradient in a forward pass to obtain each layer's parameters gradient through an operator involving *vijp* with respect to the layer's input, as well as *vjp* with respect to its parameters. Computing the input gradient in the first phase can be done in two ways: pure-forward via forward-mode differentiation, by computing the full Jacobian of just the input, which is typically much smaller than the Jacobian of the entire network; and mixed-mode, by computing the input gradient in reverse-mode. Pure-forward Moonwalk is significantly faster than full Forward differentiation, and potentially fast enough to be worth the immense memory saving over Backprop, particularly when the dimensionality of the input is small. When the input is high-dimensional, mixed-mode Moonwalk is preferred, greatly accelerating the computation compared to full Forward at the cost of more memory than pure-forward Moonwalk, but still less memory than full Backprop.

In summary, this work contributes two novel automatic differentiation methods for invertible and, more generally, submersive networks that compute full gradients in forward-mode:

- Pure-forward Moonwalk, an entirely forward-mode method that significantly reduces the time requirements of naïve forward differentiation and is the first forward-mode method to feasibly address the memory challenge of Backprop, being particularly fast when the input dimension is very small; and
- Mixed-mode Moonwalk, a variant implementing the first phase of Moonwalk in reverse-mode for further acceleration to achieve time complexity comparable with Backprop, while maintaining a smaller memory footprint than Backprop.

## 2 Related Work

**Reducing memory.** Previous studies have used checkpointing to alleviate the memory footprint of neural networks (Martens & Sutskever, 2012; Chen et al., 2016; Gruslys et al., 2016; Kumar et al., 2019; Zhao et al., 2023). This technique reduces the memory consumption in a network with $L$ layers by a factor of $\sqrt{L}$ through the selective storage of activations at intervals of $\sqrt{L}$ layers and their forward-mode recomputation between checkpoints occurs as they become needed during Backprop. This can be viewed as an equivalent network with fewer layers whose Jacobian-vector products are harder to compute, and our result applies equally to these networks as they scale up.

**Invertible architectures.** Recently, invertible (also known as reversible) architectures have gained significant attention owing to their diverse applications in reducing memory (Gomez et al., 2017; MacKay et al., 2018; Mangalam et al., 2022), enhancing learned representation (Jacobsen et al., 2018), boosting performance (Kingma & Dhariwal, 2018), and generative modeling (Dinh et al., 2014; Rezende & Mohamed, 2015). One key benefit of these architectures is the ability to avoid storing activations altogether. When the input to each layer can be computed from its output using the inverse function, several methods recompute activations backward during Backprop (Gomez et al., 2017; MacKay et al., 2018; Mangalam et al., 2022). For example, Bulo et al. (2018) replaced ReLU and batch normalization layers with invertible variants, reducing memory usage by up to 50%. Additionally, Jaderberg et al. (2017) used synthetic gradients for efficient activation storage, leveraging pre-trained gradient estimators.

**Forward propagation.** The concept of learning neural network weights in a forward fashion has been originally explored in the real-time recurrent learning (RTRL) algorithm (Williams & Zipser, 1989), similar to forward-mode AD. This idea becomes more attractive when directional derivatives (effectively, gradient projections) are employed to eliminate the additional computation time of full

forward differentiation. However, this introduces noise into the gradients (Silver et al., 2021; Baydin et al., 2022), which is coming from the random directions or imperfect gradient predictions used in directional derivatives. To address the variance of the forward gradients, Ren et al. (2022) proposed using local greedy loss functions and Fournier et al. (2023) employed local auxiliary networks as the tangent vectors, but these efforts still vastly underperform true gradients in gradient-based optimization.

## 3 Background

### 3.1 Notation

Consider a neural network $f_\theta : \mathbb{R}^n \to \mathbb{R}^k$ with $L$ layers and parameters $\theta = \{\theta_i\}_{i=1,...,L}$. Where $|\theta_i| = d_i$ is a parameter of a layer. We will denote the output of layer $i \in \{1,...,L\}$ by $x_i = f_i(x_{i-1}; \theta_i) \in \mathbb{R}^{n_i}$, where $x_0 \in \mathbb{R}^n$ is the input to the network. Let $J_\theta(x_L) = J(f_\theta(x_0))$ be the scalar loss function, whose gradient with respect to $\theta$ we wish to compute as part of a gradient-based optimization algorithm.

**Definition 1** (Submersion). *A differentiable function $f : \mathbb{R}^n \to \mathbb{R}^k$ is a submersion if its differential $\mathrm{d}\,f(x)$ is surjective for all $x \in \mathbb{R}^n$.*

Smooth submersions are useful in differential topology, where they are defined more generally for differentiable maps between differentiable manifolds, but we focus on differentiable functions between vector spaces, where the differential is simply right-multiplication by the $k \times n$ Jacobian $\partial f/\partial x$. A submersion then has $k \leq n$ and a Jacobian that is surjective (right-invertible) for all inputs. We call a neural network *submersive* if all its layers are submersions for any value of their parameters. Note that invertible networks are all submersive, because invertible layers have invertible Jacobians, but not all submersive networks are invertible.

Throughout the paper we refer to the *Jacobian-vector product*, the *vector-Jacobian product*, and the *vector-inverse-Jacobian product* as *jvp*, *vjp*, and *vijp*, respectively, and define these operators as

$$\mathrm{jvp}(f, \theta, u) = (\partial f/\partial \theta)\, u, \tag{1}$$

$$\mathrm{vjp}(f, \theta, v) = v\, (\partial f/\partial \theta)\,, \text{ and} \tag{2}$$

$$\mathrm{vijp}(f, \theta, v) = v\, (\partial f/\partial \theta)^+\,, \tag{3}$$

where $u$ is the tangent column vector, $v$ is the cotangent row vector, $(\cdot)^+$ is any right-inverse, and the Jacobian, taken here with respect to $f$'s parameters $\theta$, can instead be taken with respect to $f$'s input $x$. *jvp* and *vjp* are commonly used operators in AD frameworks, and we use their JAX implementation `jax.jvp` and `jax.vjp` (Bradbury et al., 2018). *vijp* is implemented by calling `jax.vjp` with invertible layers. For non-invertible submersions we define right inverse using SVD in appendix (Algorithm) 7

### 3.2 Forward-Mode Gradients

Forward-mode differentiation is an alternative to Backprop for computing the gradients[1]

$$\frac{\partial J}{\partial \theta_i} = \frac{\partial J}{\partial x_L} \left( \prod_{j=L}^{i+1} \frac{\partial x_j}{\partial x_{j-1}} \right) \frac{\partial x_i}{\partial \theta_i}. \tag{4}$$

While Backprop computes the product in (4) from left to right, forward-mode computes it from right to left. The suffix of the product can be computed during the forward execution of the function, such that the activations $x_i$ need not be stored, unlike in Backprop. On the other hand, while the prefixes are vectors of dimension $n$ that can be computed using *vjp*, the suffixes are matrices of dimensions $n \times d$. Where $d$ corresponds to parameter size, in general, every layer can have different $d_i$. To avoid storing these matrices in memory, they are commonly computed column-by-column using *jvp*, which increases the asymptotic time complexity of forward-mode by a factor of $\min(n, d)L$ compared to Backprop (Table 1).

---

[1]We use the notation of row-vector gradients, which are more accurately called *total derivatives*.

### 3.3 Projected Forward-Mode Gradients

Projected forward-mode gradients (Silver et al., 2021; Baydin et al., 2022) are the directional derivatives computed in a forward fashion. For a unit-length tangent vector $u \in \mathbb{R}^d$, the projection of the gradient onto $u$ has length $\text{jvp}(J, \theta, u) = \frac{\partial J}{\partial \theta} u$, and the projected vector $\frac{\partial J}{\partial \theta} u u^\intercal$ can be used as the gradient estimator. The vector $u$ is usually picked at random from a normal distribution or predicted based on past gradients and showed to be a descend direction (Silver et al., 2021; Baydin et al., 2022). Now the *jvp* can be computed recursively in forward-mode. Computing projected forward-mode gradients, then, matches the asymptotic time complexity of Backprop (section 5) but introduces some variance into the gradient estimator, which now depends on the choice of $u$.

## 4 Moonwalk

In order to benefit from the memory advantage of forward-mode gradient computation, while keeping the time complexity similar to that of Backprop and avoiding the introduction of noisy gradients through projection, we restrict our attention to the class of submersive neural networks, in which the Jacobian of each layer with respect to its input is also guaranteed to be right-invertible. We can rewrite each layer's parameter gradient of the loss as

$$g_i := \frac{\partial J}{\partial \theta_i} = \frac{\partial J}{\partial x_i} \frac{\partial x_i}{\partial \theta_i} = \frac{\partial J}{\partial x_i} \frac{\partial x_i}{\partial x_0} \left( \frac{\partial x_i}{\partial x_0} \right)^+ \frac{\partial x_i}{\partial \theta_i} \tag{5}$$

$$= \frac{\partial J}{\partial x_0} \left( \prod_{j=i}^{1} \frac{\partial x_j}{\partial x_{j-1}} \right)^+ \frac{\partial x_i}{\partial \theta_i} \tag{6}$$

$$= \frac{\partial J}{\partial x_0} \prod_{j=1}^{i} \left( \frac{\partial x_j}{\partial x_{j-1}} \right)^+ \frac{\partial x_i}{\partial \theta_i}. \tag{7}$$

Given the input gradient $\frac{\partial J}{\partial x_0}$, we can use (7) from left to right to compute the parameter gradient of each layer in forward-mode. To this end, denote the activation gradient of layer $i$ by

$$h_i := \frac{\partial J}{\partial x_i} = \frac{\partial J}{\partial x_0} \prod_{j=1}^{i} \left( \frac{\partial x_j}{\partial x_{j-1}} \right)^+, \tag{8}$$

for $1 \leq i \leq L$, and the input gradient by $h_0 := \frac{\partial J}{\partial x_0}$. Then $h_i$ satisfies the forward recursion

$$h_i = h_{i-1} \left( \frac{\partial x_i}{\partial x_{i-1}} \right)^+ = \text{vijp}(f_i, x_{i-1}, h_{i-1}), \tag{9}$$

and from (7) we have

$$g_i = h_i \frac{\partial x_i}{\partial \theta_i} = \text{vjp}(f_i, \theta_i, h_i). \tag{10}$$

Assuming that we have the input gradient $h_0$, we can construct the parameter gradient for each layer on-the-fly by the two operators in equations (9) and (10) and store only $h_i \in \mathbb{R}^{n_i}$ for the next layer's computation. The complete procedure is given in Algorithm 1 and illustrated in Figure 1c.

In the next two sections we describe the computation of $h_0$ with either forward-mode or Backprop and discuss the trade-offs of using each variant.

### 4.1 Pure-Forward Moonwalk

One way to obtain the input gradient $h_0$ is to compute it element-by-element using a projected forward-mode with a standard basis of tangent vectors. Specifically, $h_{0,i}$, the $i$-th component of $h_0$, can be computed with $\text{jvp}(J, x_0, e_i)$ with the $i$-th standard basis vector $e_i$ as the tangent vector. This method thus uses forward-mode $n$ times to construct $h_0$.

Figure 1a shows the computation flow of $h_0$ in this method. Note that, by computing $h_0$ in forward-mode, we avoid storing activations, and only store the components of $h_0$ for later computations. On the other hand, when the input dimension is large, this method takes infeasible time.

---

**Algorithm 1** Moonwalk

> **for each** gradient step with input $x_0$ **do**
>      Compute $h_0 \leftarrow \frac{\partial J}{\partial x_0}$
>      **for** $i = 1, \dots, L$ **do**
>          $x_i \leftarrow f_i(x_{i-1}; \theta_i)$
>          $h_i \leftarrow \mathrm{vijp}(f_i, x_{i-1}, h_{i-1})$
>          $g_i \leftarrow \mathrm{vjp}(f_i, \theta_i, h_i)$
>          Apply the gradient $g_i$ to $\theta_i$
>      **end for**
> **end for**

---

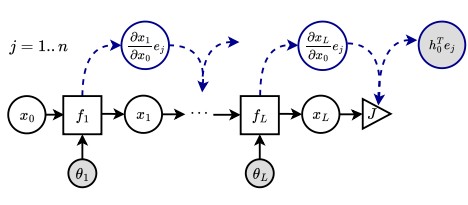

(a) Obtaining $h_0$ with the Forward gradients

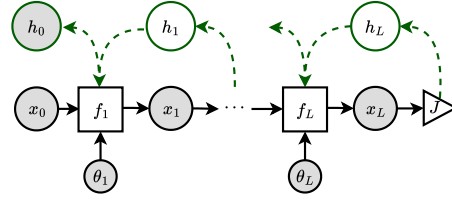

(b) Computing $h_0$ with Backprop

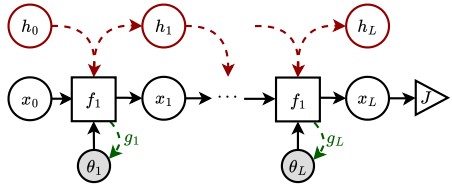

(c) Computing gradients in forward-mode given $h_0$

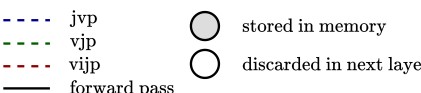

Figure 1: The computation flow diagram of Moonwalk. For comparison, diagrams for Backprop and Forward are in appendix Figure 5.

### 4.2 MIXED-MODE MOONWALK

Alternatively, we can use Backprop to compute $h_0$. While it may seem counterintuitive that a method including this phase could improve on full Backprop, the key insight is that only part of the computation graph is downstream of $x_0$. When using Backprop for $h_0$, we can therefore avoid storing activations computed from parameters $\theta_i$ independent of any $x_i$. Another consideration, more practical than principled, is that many Backprop implementations store all gradients before applying any of them, even when the parameters are layer-wise disjoint and can be updated immediately during Backprop and their gradients discarded. Here, in contrast, computing only the input gradient avoids parameter gradients altogether. In many network architectures of interest, these distinctions lead to significant memory savings in Backprop when only computing the input gradient. Figure 1b shows the flow of this procedure.

Computing $h_0$ via Backprop reduces the time complexity by a factor of up to $n$, if $d = O(n)$, compared to forward-mode (see Table 1), but may increase the memory footprint, which in the worst case can now be as high as Backprop's. However, depending on the network architecture, as analyzed in the next section and demonstrated in Section 6.2, real-world architectures are often much better than this worst case, leading to significant memory saving over Backprop. We also note that, when a checkpointed implementation of Backprop can reduce the effective number of layers, mixed-mode Moonwalk can use it as well.

## 5 COMPLEXITY ANALYSIS

While estimating the exact time and memory consumption of different methods for computing the gradients hugely depends on the choice of the network's architecture and the detailed implementation, in this section, we will provide an asymptotic analysis, in terms of the main architectural parameters, of the time and memory complexities of our methods and compare them with related previous works (Table 1). We omit all methods' linear dependence in time and memory on the mini-batch size. We analyze the computational complexity of the following methods:

1. **Backprop:** Throughout the forward pass, all (checkpointed) activations are cached, and subsequently, during a backward pass, gradients for each layer are computed using *vjp*.

2. **Forward:** During the forward pass, complete Jacobians for each layer are computed using *jvp*. In practice, a separate forward pass is used for each column to reuse memory.

3. **ProjForward:** In Projected Forward (Baydin et al., 2022), parameter gradients projected in a random or predicted direction are obtained using *jvp* concurrently with the forward pass.

4. **RevBackprop:** In Reversible Backprop (Gomez et al., 2017), no activations are stored during the forward pass. In a subsequent backward pass, the output of each layer is used to compute its input via the inverse function, as well as its parameter gradient via *vjp*.

5. **Moonwalk:** Initially, the input gradient is computed using Forward. Then parameter gradients are obtained using *vijp* and *vjp* in a second forward pass (Section 4.1).

6. **Mixed:** Similar to Moonwalk, but the input gradient is computed using Backprop, to reduce computation time at the expense of some memory impact (Section 4.2).

We evaluate time based on the standard cost of matrix multiplication, i.e. the product of their two outer dimensions and shared inner dimension, without considering optimization tricks, sparse layers, or other network structures. To evaluate memory, we define $M_{x,i}$ to be the required memory to store the necessary information to compute $\frac{\partial x_i}{\partial x_{i-1}}$, and $M_{\theta,i}$ the added memory to also compute $\frac{\partial x_i}{\partial \theta_i}$. For simplicity, we assume that these values are the same across layers and omit the layer index. We refer to memory consumption as the extra amount of memory needed to compute gradients without reflecting the memory to store the parameters or gradients themselves after computation.

**Memory complexity**: For Backprop, we have to store activations required for both input and parameter gradients for every layer, which results in $O(M_x L + M_\theta L)$ memory complexity. For Mixed, we only need to store $M_x$ for every layer in the first phase, in order to compute the input gradient $h_0$, and then we can reuse $M_\theta$ in the second phase after computing each parameter gradient, for a total memory complexity of $O(M_x L + M_\theta)$. All other methods can discard activation information after each layer, for a memory complexity of $O(M_x + M_\theta)$.

**Memory complexity with checkpointing**: In the case of Backprop with checkpointing, we will have additional memory of $O(cn)$, where $c \leq L$ is the number of checkpoints and $n$ is a bound on each layer's size. Then, during backward, we must reconstruct each block of $L/c$ layers and store activations in $O((M_x + M_\theta)L/c)$ memory. The best trade-off, obtained at $c = O(\sqrt{(M_x + M_\theta)L/n})$, is $O(\sqrt{n(M_x + M_\theta)L})$ memory. We can similarly apply checkpointing to the first phase of Mixed, which has no need to store $M_\theta$ when reconstructing from a checkpoint, for overall memory of $O(\sqrt{nM_x L} + M_\theta)$. In that case, we still prefer Mixed when $M_\theta \gg M_x$, although to a lesser extent than without checkpointing: in the extreme case that layers are so complex that we should checkpoint each one, $nL = O(M_x + M_\theta)$ and both Backprop and Mixed require $O(M_x + M_\theta)$ memory.

**Time complexity for Backprop and RevBackprop**: Backprop computation consists of computing two vector-Jacobian products in each layer, vjp$(f_i, x_{i-1}, h_i)$ and vjp$(f_i, \theta_i, h_i)$, which accounts for per-layer time complexity of $O(n^2)$ and $O(nd)$, respectively, and for a total of $O(n^2 L + ndL)$ time. RevBackprop additionally needs to evaluate the inverse function $f_i^{-1}(x_i)$, which does not impact the overall complexity in the terms we consider.

**Time complexity for Forward and ProjForward**: In Forward, each single parameter $\theta_{j,\ell}$ in layer $j$, of the total $dL$ parameters, generates a pass to compute its gradient, in which we compute jvp$(f_i, x_{i-1}, \frac{\partial x_{i-1}}{\partial \theta_{j,\ell}})$ which has a complexity of $O(n^2)$ in each layer $i > j$ for $L$ layers, for a total of $O(n^2 dL^2)$ time. ProjForward with tangent $u = \{u_i\}_{i=1,\dots,L}$ is similar to Forward with just a single

Table 1: Asymptotic complexity and key characteristics of pure-forward Moonwalk, its mixed-mode variant, and four existing methods, analyzed in Section 5. **Stable:** numerically stable; **Forward:** computes gradients only during forward passes; **Submersive:** applicable to submersive networks.

| Method | Time | Memory | Stochastic | Stable | Forward | Submersive |
|---|---|---|---|---|---|---|
| Backprop | $O(n^2L + ndL)$ | $O(M_xL + M_\theta L)$ | ✗ | ✓ | ✗ | ✓ |
| Backprop + checkpoint | $O(n^2L + ndL)$ | $O(\sqrt{n(M_x + M_\theta)L})$ | ✗ | ✓ | ✗ | ✓ |
| Forward | $O(n^2dL^2)$ | $O(M_x + M_\theta)$ | ✗ | ✓ | ✓ | ✓ |
| ProjForward | $O(n^2L + ndL)$ | $O(M_x + M_\theta)$ | ✓ | ✓ | ✓ | ✓ |
| RevBackprop | $O(n^2L + ndL)$ | $O(M_x + M_\theta)$ | ✗ | ✗ | ✗ | ✗ |
| Moonwalk | $O(n^3L + ndL)$ | $O(M_x + M_\theta)$ | ✗ | ✓ | ✓ | ✓ |
| Mixed | $O(n^2L + ndL)$ | $O(M_xL + M_\theta)$ | ✗ | ✓ | ✗ | ✓ |
| Mixed + checkpoint | $O(n^2L + ndL)$ | $O(\sqrt{nM_xL} + M_\theta)$ | ✗ | ✓ | ✗ | ✓ |

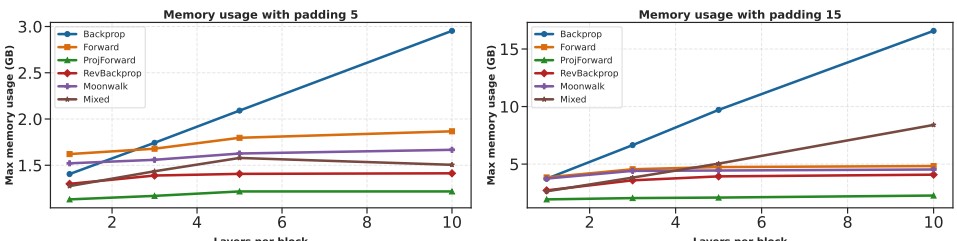

Figure 2: Maximum allocated memory during training with a different number of layers per block. (a) Input is padded from 32x32x5 to 32x32x8, (b) Input is padded from 32x32x3 to 32x32x18.

pass taking $O(n^2L)$ time, but in each layer accumulating $\text{jvp}(f_i, \theta_i, u_i)$, for a total of $O(n^2L + ndL)$ time, which coincides with the time complexity of Backprop.

**Time complexity for Moonwalk and Mixed**: The first phase of pure-forward Moonwalk computes $\text{jvp}(f_i, x_{i-1}, \frac{\partial x_{i-1}}{\partial x_0} e_\ell)$ in each layer for each input element $\ell$, for a total time complexity of $O(n^3L)$. The second phase computes $\text{vijp}(f_i, x_{i-1}, h_{i-1})$ and $\text{vjp}(f_i, \theta_i, h_i)$ in each layer for $O(n^2L + ndL)$ more, and a total of $O(n^3L + ndL)$ time. Mixed-mode replaces the first phase with Backprop for just the input gradient, incurring time complexity $O(n^2L)$, in addition to $O(n^2L + ndL)$ for the same second phase as in Moonwalk, for a total of $O(n^2L + ndL)$ time complexity.

Table 1 summarizes the order of growth of time and memory in the methods we compare, and the next section evaluates them empirically.

# 6 EXPERIMENTS

## 6.1 EXPERIMENTAL SETUP

**Model and architecture.** We adopt a modified RevNet architecture (Gomez et al., 2017) with three blocks. Each block of the network consists of coupling layers that partition the input into two subsets $x_1$ and $x_2$, and output

$$\begin{aligned} y_1 &= x_1 \\ y_2 &= x_2 + \mathcal{F}(x_1). \end{aligned} \tag{11}$$

The function $\mathcal{F}$ can be any arbitrary function, not necessarily invertible. In our case, it is represented by a series of convolutional layers, each followed by a ReLu activation (Agarap, 2018) and BatchNorm (Ioffe & Szegedy, 2015) layers. The input to each layer can be reconstructed from it is output, without

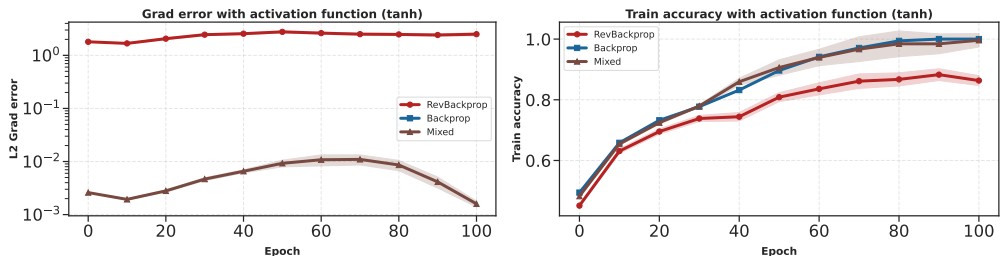

Figure 3: (a) Gradient error with activation function (tanh) on RevBackprop, Backprop, and Mixed over 100 epochs, averaged over 20 runs. (b) Train accuracy of three models trained with RevBackprop, Backprop, and Mixed gradient methods for 100 epochs, averaged over 20 runs.

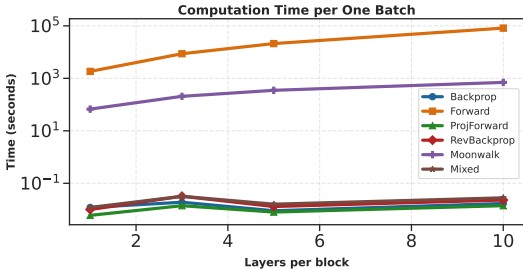

Figure 4: Log-scale computation time per one batch for a different number of layers per block

the need to cache activations, by

$$
\begin{aligned}
x_1 &= y_1 \\
x_2 &= y_2 - \mathcal{F}(x_1).
\end{aligned}
\tag{12}
$$

In our experiments, the network consists of three sequential blocks as depicted in 6. The number of layers in each block is set in each experiment to one of $\{1, 3, 5, 10\}$. Following each block, we apply an invertible down-sampling operator $\psi$, as proposed by (Jacobsen et al., 2018). The operator $\psi$ effectively divides the spatial dimensions into the channel dimension, resulting in a fourfold increase in the feature-space dimension for each successive block.

First, we demonstrate that Forward is markedly more memory efficient than Backprop by measuring the memory consumption of the network during the training of each minibatch. Second, we show that Moonwalk is substantially faster than the Forward. By imposing constraints on the network architecture, we substantially reduce the overall gradient computation time. Third, we illustrate that Moonwalk can be combined with Backprop, further reducing the computation time while preserving significant memory savings. Finally, we establish that Mixed exhibits greater robustness compared to RevBackprop.

## 6.2 MEMORY REDUCTION

For our first experiment, we pad the input size from 32x32x5 to 32x32x8. The results summarized in Figure 7 suggest that the memory consumption of Backprop increases linearly with the number of layers per block. Backprop necessitates the most substantial memory allocation among the compared approaches, resulting in a twofold disparity at ten layers per block for Moonwalk, Mixed, and RevBackprop. For ProjForward, the disparity is even larger at 2.4 times. The disparity growth for Forward is a bit slower due to its initial compilation overhead. Additionally, it is noteworthy to observe an initial compilation gap, resulting in increased memory utilization when the architecture comprises only one layer per block. The memory footprint in Backprop scales with layer size, while Forward memory growth is confined solely to the number of layers. This reduction is achieved by discarding activations that are no longer needed for the forward pass. The resulting memory footprint for Forward consists only of the memory allocated for the storage of gradient updates until they are applied to the model parameters. Furthermore, we evaluated methods on another type of network,

where we padded the input from 32x32x3 to the size of 32x32x18. In such cases, memory is mostly dominated by $M_x L$. Our demonstration in Figure 7 reveals that such architecture choice results in an expanded disparity between Backprop and forward methods. Notably, Mixed experiences an escalation in memory consumption as its allocation begins to be dominated by the activation preserved for gradient computation of the input.

### 6.3 Computation Time

While Forward showcases a substantial reduction in memory usage, it encounters challenges in terms of computation time (Figure 4). The approach demands a considerable amount of computation, rendering it impractical in many scenarios. In contrast, Moonwalk significantly reduces computation time, achieving efficiency up to several orders of magnitude compared to Forward. We evaluate all methods using the same architecture, incorporating varying numbers of layers per block.

The results, illustrated in Figure 4, accentuate the notable efficiency gains achieved by Moonwalk. In the case of six layers, Forward takes 1839 seconds per batch, whereas Moonwalk completes the task in 67 seconds, representing a substantial 27-fold reduction in computing time. Extending the analysis to a model with 60 layers, Forward demands more than 80000 seconds, while our method accomplishes the same task in 700 seconds, manifesting a remarkable time reduction factor of 110.

These outcomes underscore the significant impact on training duration, revealing that employing Moonwalk for training a full model on the CIFAR-10 dataset requires approximately five days—an appreciable improvement over Forward, which necessitates around 300 days. By using Mixed Moonwalk we can further reduce training of the full model to a few hours.

### 6.4 Time-Memory trade-off

While achieving noteworthy time reduction using Moonwalk, we can further optimize efficiency by precomputing the gradients of the input, denoted as $h_0$, in our method using backpropagation. Although the memory reduction in this approach is constrained by the memory required for activations, it significantly decreases overall memory consumption compared to Backprop. Two pivotal components contribute to this reduction:

**Elimination of Activations Storage**: We refrain from storing activations utilized in computing the gradients of the weights $M_\theta L$. However, it is important to note that in this scenario, we are still bound by the activations necessary for computing $h_0$. In instances where input is padded to 32x32x8 (see Figure 7), we do not incur a memory disadvantage compared to alternative methods. In scenarios featuring larger input size 32x32x18, as depicted in Figure 7, a noticeable increase in memory usage over Forward becomes apparent.

**Sequential Memory Usage**: There is no longer a need to store gradients and activations simultaneously, as Backprop does. After precomputing $h_0$, we discard all activations, freeing up memory for gradient computations.

In Figure 4, Mixed is only about twice as slow as Backprop. The primary advantage of this method lies in its substantial time reduction compared to Forward.

### 6.5 Submersive Layers

**Definition 2** (Linear Submersive Layer). *A linear submersive layer is a fully connected linear layer* $\mathbf{y} = \mathbf{W}\mathbf{x}$*, where* $\mathbf{W} \in \mathbb{R}^{k \times n}$ *with* $k \leq n$*. Additionally, there exists a matrix* $\mathbf{W}^+ \in \mathbb{R}^{n \times k}$ *such that* $\mathbf{W}\mathbf{W}^+ = \mathbf{I} \in \mathbb{R}^{k \times k}$*.*

**Definition 3** (Stable Linear Submersive Layer). *A stable linear submersive layer is a linear submersive layer where the weight matrix* $\mathbf{W} \in \mathbb{R}^{k \times n}$ *is constrained to be upper triangular with ones on its main diagonal.*

**Definition 4** (Stable Submersive 1D Convolution). *A 1D convolution with the first weight fixed is a convolutional operation defined as*

$$\mathbf{y}[i] = \sum_{k=0}^{K-1} w_k \mathbf{x}[i+k],$$

*where $\mathbf{x}$ is the input, $\mathbf{y}$ is the output, $w_k$ are the weights of the convolution kernel of size $K$. The constraint $w_0 = 1$ is imposed on the first weight of the kernel.*

One of the key highlights of our method is its ability to work effectively with submersive networks. Here, we expand further on this class of networks. The simplest case of a submersion layer is a linear layer (Definition 3) with contracting dimensions, i.e., $f(x) : \mathbb{R}^n \to \mathbb{R}^k$, where $k \leq n$. For this layer to qualify as a submersion, we require its Jacobian to be right-invertible. To ensure numerical stability, we propose parametrizing such layers with ones on their main diagonal. This parametrization enables a stable inversion algorithm based on Gaussian Elimination (Algorithm 6) or SVD (Algorithm 7).

## 6.6 Experiments with Submersive Layers

To demonstrate Moonwalk's applicability to submersive layers, we design an architecture composed of submersive layers, where each linear layer is followed by a LeakyReLU activation function. We highlight the key differences in computing $\nabla X$ between backpropagation and Moonwalk in Algorithm 2 and Algorithm 3.

The primary distinction lies in memory requirements. During the computation of $\nabla X$, Moonwalk eliminates the need to store intermediate activations ($z_2, x$) and gradients ($\nabla W_1, \nabla W_2$). Instead, it only requires storing the LeakyReLU gradient (LeakyReLUGrad), represented as a binary matrix of signs. This approach significantly reduces memory usage, requiring approximately 16 to 32 times fewer bytes compared to storing activations in FP16 or FP32 precision.

Another critical difference is that Moonwalk requires $\nabla X$ to compute $\nabla W_1$ later, as outlined in Algorithm 4. Finally, to validate the effectiveness of our method, we train a submersive network on the CIFAR-10 dataset (Krizhevsky et al.). We did not explicitly measure memory consumption for this experiment, as the overhead introduced by linear layers outweighs the memory differences between the methods. However, for convolutional layers, the memory benefit of Moonwalk becomes more apparent, as the gradients of the weights occupy roughly the same amount of memory as activations.

To further support our approach, we include a plot of gradient error (Figure 8), demonstrating that the weight matrices converge over a few examples, even with a batch size of 1.

## 7 Conclusion and Future work

The efficiency of memory utilization in Forward surpasses that of Backprop, thereby alleviating the challenges associated with activation storage constraints. However, this enhanced memory efficiency comes at a considerable computational cost, rendering Forward computationally impractical. To address this limitation, we augmented Forward processing through the incorporation of invertible networks. Our experimental findings reveal that our proposed method, Moonwalk, exhibits a marked acceleration, surpassing Forward by several orders of magnitude. Furthermore, we proposed an approach to mitigate the computational overhead associated with our method. By integrating Backprop for the computation of input gradients, we achieved a substantial acceleration of overall gradient computation.

In this paper, we restricted our attention to the class of invertible and submersive layers in order to compute the gradients for the next layer given the previous one. Future work would study the effect of using projected forward-mode gradients instead of the full forward-mode. Also, one could investigate the class of neural networks composed of locally invertible layers to shed light on the applicability of Moonwalk for a larger family of architectures.

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

# 8 APPENDIX

## 8.1 DATA AND HYPER-PARAMETERS.

We implement all methods using the JAX framework (Bradbury et al., 2018) and conduct experiments on training a neural network on the CIFAR-10 dataset (Krizhevsky et al.). The experiments are performed on an RTX A4500 GPU with a batch size of 512. Computation time is measured after jit-compilation is completed. Memory consumption is tracked using the nvidia-smi utility every second, with the maximum value recorded. Data preprocessing involves padding with zeros in the channel dimension, expanding the input dimension from 32x32x3 to to one of {32x32x18,32x32x8}. We use the Adam optimizer (Kingma & Ba, 2017) with a learning rate of $10^{-3}$, and we use an exponential decay scheduler unless specified otherwise.

## 8.2 NUMERICAL STABILITY

It is worth noting that RevBackprop (MacKay et al., 2018), while requiring a bijective network with similar computation time, encounters challenges in terms of numerical stability. As demonstrated by (Gomez et al., 2017) and (Jacobsen et al., 2018), invertible architectures with finite precision encounter numerical instability issues, leading to divergence from true gradients and convergence to a different local minimum than Backprop. In this study, we highlight that RevBackprop fails to converge when an activation function is added to the output of the layer. Employing the same network and dataloader for simultaneous training, Fig 3 illustrates that both Backprop and Mixed achieve perfect accuracy, while RevBackprop fails to converge. This divergence is attributed to the error accumulating with each update.

Fig 3 displays the gradient error estimation between the algorithm's gradients and oracle gradients at each step using the same parameters. The experiment reveals the accumulating error causing the network weights to drift away from true gradients. Mixed exhibits more stable convergence due to lower gradient error. Such disparity is a result of different approaches involved in gradient computation. In order to compute the gradients, RevBackprop requires computing the inverse of the function, which might be extremely unstable, as in the case with a tanh activation function. As opposed to RevBackprop, Moonwalk and Mixed methods only require computing inverse-vector Jacobian product and avoid computation of the inverse function itself. In some cases, as we have shown with tanh, this tends to be the more stable approach to computing gradients. The impact of "gradient drifting" in the reversible method becomes noticeable after 30 epochs.

## 8.3 SUBMERSIVE NETWORKS

---

**Algorithm 2** Backpropagation

---

**Require:** $W_2, W_1, x$
**Ensure:** $err, \nabla W_1, \nabla W_2$
  1: **Store** $x$ {Forward Part}
  2: $z_1 \leftarrow \text{linear\_layer}(W_1, x)$
  3: $z_2 \leftarrow \text{leaky\_relu}(z_1)$
  4: **Store** $z_2$
  5: **Store** LeakyReluGrad $\leftarrow \text{where}(z_2 > 0, 0.01, 1)$
  6: $z_3 \leftarrow \text{linear\_layer}(W_2, z_2)$
  7: $y\_hat \leftarrow \text{sum}(z_3)$ {Predicted output}
  8: $err \leftarrow -2 \cdot (y - y\_hat)$ {Backward Part}
  9: $\nabla W_2 \leftarrow (err \cdot \text{ones\_like}(z_3)) \cdot z_2^\top$
 10: **Discard** $z_2$
 11: $\nabla W_1 \leftarrow err \cdot (W_2 \cdot \text{signs\_for\_grads}) \cdot x^\top$
 12: **Discard** $x$
 13: **Discard** signs_for_grads
 14: **return** $err, \nabla W_1, \nabla W_2$

---

**Algorithm 3** Moonwalk $\nabla X$

---

**Require:** $W_2, W_1, x$
**Ensure:** Gradient of the input $(\nabla X)$
1: $z_1 \leftarrow$ linear_layer$(W_1, x)$
2: $z_2 \leftarrow$ leaky_relu$(z_1)$
3: **Store** LeakyReluGrad $\leftarrow$ where$(z_2 > 0, 0.01, 1)$
4: $z_3 \leftarrow$ linear_layer$(W_2, z_2)$
5: $y\_hat \leftarrow$ sum$(z_3)$ {Predicted output}
6: err $\leftarrow -2 \cdot (y - y\_hat)$ {Backward Part}
7: res $\leftarrow (W_2) \cdot (W_1 \cdot$ signs_for_grads$)$
8: **Discard** LeakyReluGrad
9: **return** err $\cdot$ res.sum$(0)$

---

**Algorithm 4** Moonwalk $\nabla W_1$

---

**Require:** $W_2, W_1, x, \nabla X$
**Ensure:** Gradient with respect to $W_1$
1: inv_jacobian $\leftarrow$ inverse_upper$(W_1)$
2: $z \leftarrow \nabla X \cdot$ inv_jacobian
3: **return** $z \cdot x$

---

**Algorithm 5** Moonwalk $\nabla W_2$

---

**Require:** $W_2, W_1, x, \nabla X$
**Ensure:** Gradient with respect to $W_2$
1: inv_jacobian $\leftarrow$ inverse_upper$(W_1)$
2: $z \leftarrow \nabla X \cdot$ inv_jacobian
3: $z_2 \leftarrow$ linear_layer$(W_1, x)$
4: $z_2 \leftarrow$ leaky_relu$(z_2)$
5: LeakyReluGrad $\leftarrow$ where$(z_2 > 0, 0.01, 1)$
6: $z \leftarrow z/$LeakyReluGrad$^\top$
7: inv_jacobian $\leftarrow$ inverse_upper$(W_2)^\top$
8: $z \leftarrow z \cdot$ inv_jacobian
9: **return** $z \cdot z_2$

---

**Algorithm 6** Calculation of Right Inverse of Upper Triangular Matrix $A$

---

**Require:** $A \in \mathbb{R}^{k \times n}$ (upper triangular matrix), $k \leq n$
**Ensure:** Right inverse matrix $B \in \mathbb{R}^{n \times k}$
1: Initialize $B \leftarrow \mathbf{0}_{n \times k}$
2: $I_k \leftarrow$ Identity matrix of size $k \times k$
3: **for** $i = k - 1$ **to** $0$ **do**
4:    $b \leftarrow I_k[i, :]$ {Current row of identity matrix}
5:    **for** $j = i + 1$ **to** $n - 1$ **do**
6:       $b \leftarrow b - A[i, j] \cdot B[j, :]$ {Subtract contribution of already solved rows}
7:    **end for**
8:    $B[i, :] \leftarrow b/A[i, i]$ {Solve for the current row of $B$}
9: **end for**
10: **return** $B$

---

**Algorithm 7** Computation of the Pseudoinverse using SVD

---

**Require:** $A \in \mathbb{R}^{k \times n}$ (matrix to compute the pseudoinverse)
**Ensure:** Pseudoinverse of $A$
1: Perform SVD: $U, S, V \leftarrow$ svd$(A)$
2: **return** $V^T S U^T$

### 8.4 Implementation

We implement all methods using JAX Bradbury et al. (2018). Forward involves applying the `jax.jvp` operator sequentially at each layer, with the number of applications corresponding to the number of parameters in that layer. For each layer of the network, we define both forward and inverse functions. To enhance computation speed, we flatten all layer parameters and create an identity matrix used for projection. Utilizing a batched function map accelerates computation, although it comes at the cost of an increased overall memory footprint for forward-based methods. An alternative to mitigate the increased memory footprint is to compute the `jax.jvp` product one by one without creating a large identity matrix. While this approach reduces the memory footprint, it also significantly increases compilation time and affects overall computation, prohibiting the use of `jax.vmap` in such scenarios. The depicted performance of Mixed in the following subsections is not optimized, though it is already only twice as slow compared to Backprop. Some of the factors, including data transfer between compiled and non-compiled environments, significantly affect its performance. In theory, it might be possible to make it faster than Reversible.

### 8.5 Stochasticity of ProjForward

ProjForward demonstrates the most conservative memory usage; however, it stands as the sole method not generating true gradients. Instead, it introduces variance in its estimation owing to the adoption of an arbitrary random distribution for projecting the parameter space. However, it has been substantiated to be advantageous in specific scenarios, particularly when a reliable estimate for the gradient direction is available (Silver et al., 2021; Baydin et al., 2022; Fournier et al., 2023).

### 8.6 Architecture

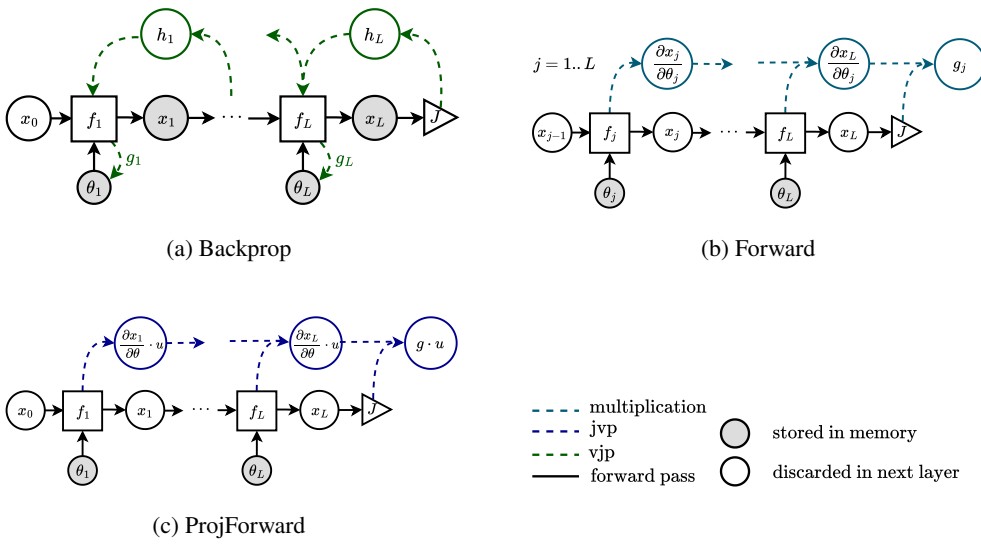

(a) Backprop

(b) Forward

(c) ProjForward

Figure 5: The computation flow diagram of Backprop, Forward, and ProjForward.

We present the architecture in Figure 6. The network comprises three main blocks (depicted in green) and several sub-blocks, with each main block containing approximately 3 to 15 sub-blocks. The input to the network is zero-padded to upscale the initial data. Each layer is constructed using an affine coupling layer parameterized by stacked convolutional layers. With the following number of channels 16,32,6 and with the kernel size of (3,3). Where each layer is followed by a ReLu activation and BatchNorm.

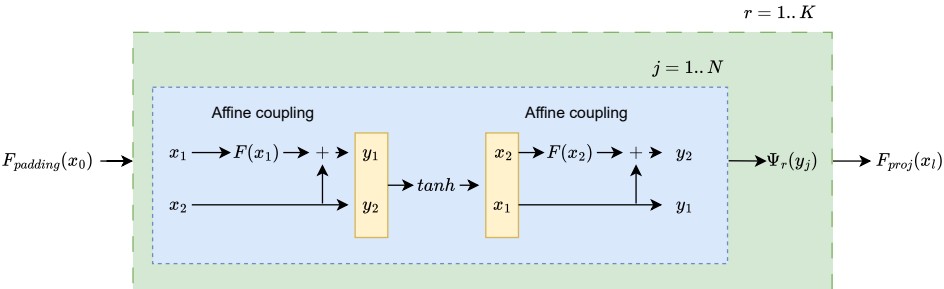

Figure 6: This figure illustrates the architecture of the model. The entire network consists of $NK$ blocks. Each inner block (denoted in blue) is followed by $\Psi(x)$ is a downscaling operator, following the definition in Jacobsen et al. (2018). Yellow blocks represent concatenation operations. $F$ denotes a block of stacked, non-invertible convolutional layers. The function $F_{padding}$ applies zero padding to the input. The final projection is performed by the layer $F_{proj}$.

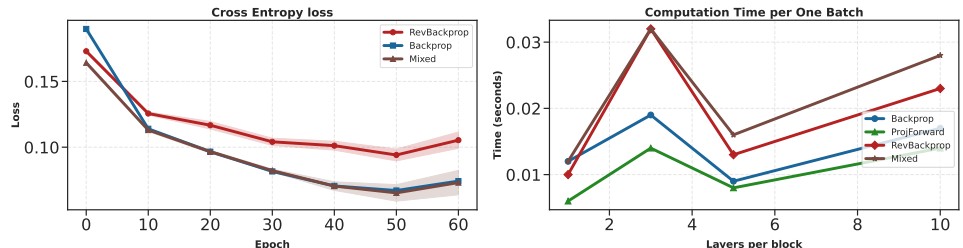

Figure 7: (a) Cross Entropy loss for different gradient methods with tanh activation over 60 epochs. (b) Computation time per batch for a different number of layers per block.

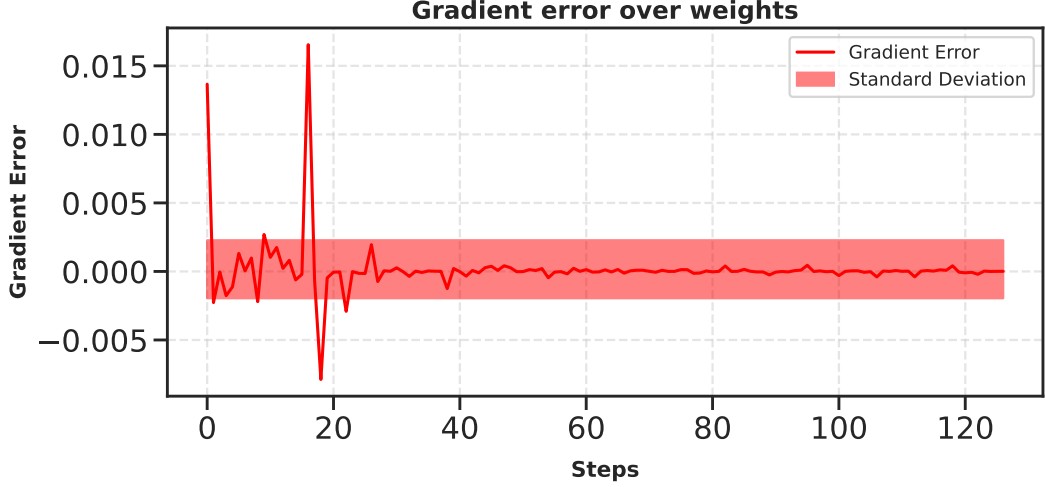

Figure 8: Gradient error $\theta_i$ for training a submersive network with linear layers and LeakyRelu with batch size 1.

