# OpenReview forum: "Moonwalk: Inverse-Forward Differentiation"
_ICLR.cc/2025/Conference — Submitted to ICLR 2025_

### Official Review · Reviewer_9rVw · 2024-10-31

**Soundness:** 2
**Presentation:** 3
**Contribution:** 2
**Rating:** 3
**Confidence:** 4

**Summary:**

This article presents an alternative to backpropagation named Moonwalk. It first computes the loss gradient w.r.t the input, either using forward-mode automatic differentiation or a standard backward pass. Then, the parameter gradients are computed during a forward pass, using reversible (or submersive) layers to compute the vector-inverse-jacobian of each. The authors implement their method in a RevNet, and compare its theoretical and practical memory and time overheads. They also experimented on the stability of the standard RevBackprop approach to reversible networks.

**Strengths:**

- This method is novel and allows for an exact forward-mode gradient computation, as soon as the input gradient is computed.
- In the case of pure-forward Moonwalk, the method improves over standard Forward-mode AD, but not necessarily over ProjForward, depending on its convergence.
- The paper is well-written and clear.

**Weaknesses:**

- The practicality of the method is very limited and over-claimed for the Mixed approach. RevBackprop is more effective on all points (see next weakness regarding the stability of RevBackprop). The use of a big O notation to hide the (approximately) doubling of the time complexity in Moonwalk is not acceptable: for almost all methods (except Forward and Forward-Moonwalk), the only difference in execution time is up to a constant, which matters. I found it surprising to claim that RevBackprop is not applicable to submersive networks, considering that all invertible networks are submersive (line 130), and the authors never give an example of a submersive network that is not invertible, considering only a RevNet. Since this is the only real advantage over RevBackprop, it requires more justification and a real network example.
- I disagree with the authors over their analysis of stability in RevBackprop. They only provide a single seemingly hand-picked example of numerical instability of RevBackprop when adding a tanh activation function. This is a very rare activation, not used in RevNets or Transformer-based models like Revformers. Other works like i-revnet et RevVit showed that stability was a non-issue and that the approximation error was equal to $10^{-6}$ in the worst cases. Without more precise examples, this example is not a demonstration of the instability of RevBackprop. Furthermore, it remains to be made clearer why computing the vector-inverse-jacobian should be more stable than the inverse function directly used in RevBackprop, since it also uses the inverse function, as explained in line 141.
- References to activation checkpointing need to be updated with more recent ones which are more general and relevant, such as [1,2].
- Despite the improvement over standard Forward-mode AD, the method is not compared in practice to the convergence of ProjForward, making it hard to choose one over the other.
- The method is limited to reversible architectures (and submersive ones), where RevBackprop is already available.



[1] Efficient rematerialization for deep networks. NeurIPS 2019, Kumar et al.
[2] Rockmate: an efficient, fast, automatic and generic tool for re-materialization in pytorch. ICML 2023, Zhao et al.

**Questions:**

**Questions**
- The possible differences between $M_x$ and $M_θ$ are unclear in the text. What are these values in some standard layers? ($W$ and $X$ for a linear layer for instance). This makes it very unclear if one value can overshadow the other in practice.
- Have the authors considered doing ProjForward to compute the input gradient rather than computing it explicitly?
- line 276: why omit it when  $M_θ$ (for a linear layer) will depend on the batch size value but not $M_x$?

**Minor details**
- Wrong use of citet/citep on some occasions (line 500 for instance).
- Figures 2-4 captions are hard to read. It is also very hard to compare the non-forward methods in Figure 4.
- The caption of Figure 3a is wrong, and the one of Figure 4 lacks a dot.
- line 176 misses a space.
- The use of the term "Pure-Forward Moonwalk" or just "Moonwalk" varies during the paper.

---

> ### Author Response · Authors · 2024-11-26
>
> We thank the reviewer for their constructive feedback and valuable suggestions, which helped us identify areas for clarification and improvement
>
> 1. "The practicality of the method is very limited"
>
> We thank the reviewer for his comment. We would like to add a clarification that all invertible networks are submersive but the opposite is not true. Not all submersive networks are invertible. We added an algorithm with an architecture example to show that we can train a submersive network, which is not invertible. In the case of Linear layers that reduce dimensionality, they are not invertible, but submersive, we provide an algorithm and code snapshots to train such networks. We added more justification, but the main point is that we can use Mixed Moonwalk with linear layers and convolutions, whereas reversible can not operate under these constraints.
> We would clarify that the main point of our work is to show a novel method for computing gradients. First, it allows us to train the network where Reversible fails to do so. We added Algorithms 2 and 3 to showcase the difference between Mixed and Backprop.
> We would also to clarify that big O notation to show that the theoretical properties of Moonwalk are similar to backpropagation, whereas optimal performance is heavily dependent on optimization.
>
> 2. "I disagree with the authors over their analysis of stability in RevBackprop"
>
> Thank you for raising this point, we do agree that evaluating one activation function is not sufficient to show a comparison between methods. We would like to highlight a few things:
>
> There is a line of work [1] that showcases where reversible networks fail with some coefficients. We did not investigate this particular example, but hypothesize, that moonwalk can solve this issue. We will add in the updated draft more experiments similar to [1].
>
> 3. References to activation checkpointing need to be updated with more recent ones which are more general and relevant, such as [1,2].
> Thank you, we updated the current draft
>
> 4. Despite the improvement over standard Forward-mode AD, the method is not compared in practice to the convergence of ProjForward, making it hard to choose one over the other.
>
> We would like to highlight that ProjForward does not produce true gradients, and in all experiments, it failed to converge.
>
> 5. The method is limited to reversible architectures (and submersive ones), where RevBackprop is already available.
>
> Please, see our point about extension to the subclass of non-invertible subversive networks. (Linear layers and Convolutions)
>
> Questions
> 1. “The possible differences between “
>
> Thank you for the comment! An example could be shown from algorithm 2 when we need to store For M_x only signs for grads, but for M_\theta we also need activation itself.
>
> 2. “Have the authors considered doing ProjForward to compute the input gradient rather than computing it explicitly”
>
> Yes, we did some experiments, without any success at this point. We believe that if we estimate it with multiple directions then when we multiply with inverse matrix the error would quickly grow. We think that more constraints are needed in order to solve this problem. In general, we think that this idea might be another excellent point to prefer moonwalk instead of backprop/reversible.
>
> 3. line 276: why omit it when
> Thanks for pointing out. Both M_x and M_\theta would depend on batch_size. Please, refer to algorithms 2 and 3, in case of adding Batch size, we will have to store N * signs_fo_grads (Which corresponds to M_x), and M_\theta would correspond to variables that we have to store for algorithm 3. In general, adding batch size as an additional parameter would benefit Moonwalk more than backpropagation.
>
> Minor details
> Updated in the new version.
>
> References:
> [1] Liao, Make Pre-trained Model Reversible: From Parameter to Memory Efficient Fine-Tuning

---

> > ### Comment · Reviewer_9rVw · 2024-11-27
> >
> > We thank the authors for their response.
> >
> > **Practicality** Indeed, a network as proposed composed of for instance linear layers with decreasing dimensionality can be used with Moonwalk but not with Reversiblity. Still, this is a very limited case, and most modern architectures do not follow this type of architecture, but one composed of residual blocks; this is the way ResNets and Transformers have been adapted into reversible networks for instance. In these cases, it will always be possible to adapt these networks with residual connections in reversible networks. This seems to limit the practicality of Moonwalk to small networks without residual connections. Still, I agree with the authors that some networks like this simple MLP are not adaptable. Appendix 8.3 necessitates some accompanying text to better explain the new Algorithms given, at least describing simply the architecture considered for instance. Why is "inverse_upper" used? The weights $W_i$ have no reason to be upper triangular if I'm not mistaken. What is the point of Algo 7 if it is not used here?
> >
> > I still disagree strongly with the use of the big O notation. It hides an almost doubling of time execution, which is not trivial in practice, and is precisely the interest of the method proposed.
> >
> > **Stability** The reference [1] shows that the numerical stability issues that can occur are not due to the computation of the inverse of the function itself, which is exactly the same function used in the forward pass; but due to the potential magnitude of the scaling factors used in the residual connection. In all standard networks like ResNets, reformers or RevViT, these factors are equal to $1$. Thus, they measure a negligible error of $10^{−8}$ only. I am not seeing the "experiments similar to [1]." discussed by the authors if I'm not mistaken. Without these, I am not convinced by the reasoning of the authors.
> >
> > **4** Thank you, this should be indicated in the paper.
> >
> > **Q1/3** Thank you for these precisions.
> >
> > **Q2** This seems logical considering the high variance of the ProjForward estimator, thank you. Although I do not understand the points of the authors regarding the inverse matrices and the comparison with (rev/)backpropagation.

---

### Official Review · Reviewer_Gc2t · 2024-11-01

**Soundness:** 3
**Presentation:** 3
**Contribution:** 3
**Rating:** 6
**Confidence:** 3

**Summary:**

The authors introduce Moonwalk, an algorithm for computing the gradient of an invertible network (or more generally submersive networks). Compared to backpropagation, it does not require storing intermediate hidden states in memory, thus allowing for a more memory-efficient training, at the cost of an increased computation time. A variation of Moonwalk is also proposed to have the same time complexity as backpropagation but still with noticeable memory savings.

**Strengths:**

- The proposed algorithm is mathematically founded, and seems intuitive and natural.
- Moonwalk, and its Mixed version, are new interesting algorithms that propose different time/memory complexity tradeoffs from existing alternatives to backpropagation.
- The time and memory analysis is thorough, and is done for the proposed algorithms and several other existing algorithms.
- Empirical benchmarks on RevNet models validate the claims, showing noticeably less memory usage compared to backpropagation.

**Weaknesses:**

1. The main issue I find in Moonwalk is computing the gradient w.r.t. the input, which is very expensive. As seen in Fig. 4, it is a couple orders of magnitude slower than backpropagation. This is extremely unpractical. However, the authors are aware of this limitation and propose the Mixed variant to mitigate it, which I find much more convincing.
2. Unlike what is stated in the abstract ("Finally, we showcase the robustness of our method across several architecture choices."), the algorithms are only tested on RevNet with 3 blocks. Only the number of layers in the blocks, the number of input channels, and the activation between blocks, are changed. I could be nice to see Moonwalk work on other inversible architectures, which would in particular make the time and memory benchmarks more convincing.
3. The algorithm is only applicable to very specific architectures, which are rarely used in practice. But this is only a minor weakness, as the use of invertible networks could actually be motivated by algorithms like Moonwalk.
4. I believe the captions must be above the tables according to the ICLR template.

**Questions:**

5. While I understand that the paper focusses on exact computation of the gradients, it would be a great addition to discuss more about estimations (like the ProjForward algorithm). There are for instance the forward-only algorithms (Forward-Forward, DFA, PEPITA…). In particular when computing the gradient wrt the input, it seems natural to try to estimate it as in ProjForward using vjp with $k$ random directions. Have you thought that or tried it?
6. Although it seems nice to extend the applicability of Moonwalk to a larger class of functions, I find it hard to get intuition of what this changes in the context of deep learning models. Do you have examples of layers which would be submersive but non invertible?

I remain open to discussion and may improve my grade in the future.

---

> ### Author Response · Authors · 2024-11-26
>
> Weaknesses:
>
> 1. The main issue I find in Moonwalk is computing the gradient w.r.t. the input, which is very expensive. As seen in Fig. 4, it is a couple orders of magnitude slower than backpropagation. This is extremely unpractical. However, the authors are aware of this limitation and propose the Mixed variant to mitigate it, which I find much more convincing.
>
> Thank you for your point. We added algorithms 2 and 3 to show that our method in general is more efficient than backprop and works on submersive networks, whereas reversible is not compatible with such architectures. We showcase that we only need to store signs of gradients rather than the entire variables
>
> 2. Unlike what is stated in the abstract ("Finally, we showcase the robustness of our method across several architecture choices."), the algorithms are only tested on RevNet with 3 blocks. Only the number of layers in the blocks, the number of input channels, and the activation between blocks, are changed. I could be nice to see Moonwalk work on other inversible architectures, which would in particular make the time and memory benchmarks more convincing.
>
> Thank you for the comment! We added algorithm 2 and code snapshots to show that our method works on submersive networks with Linear Layers and 1d Convolutions with Code examples. We would also to highlight that reversible - backprop can’t work with such architectures and we are more memory efficient than backprop in such scenarios.
>
> 3. The algorithm is only applicable to very specific architectures, which are rarely used in practice. However this is only a minor weakness, as the use of invertible networks could actually be motivated by algorithms like Moonwalk.
>
> We do agree, but as we show almost any Linear layer with an output size lesser than the input could be right invertible with Gaussian elimination (We also added an algorithm) which is more efficient than using SVD. Please, refer to the updated version of the manuscript where we included new types of layers.
>
> 4. I believe the captions must be above the tables according to the ICLR template.
>
> Thank you, we updated the tables.
>
> Questions:
>
> 1. While I understand that the paper focusses on exact computation of the gradients, it would be a great addition to discuss more about estimations (like the ProjForward algorithm). There are for instance the forward-only algorithms (Forward-Forward, DFA, PEPITA…). In particular when computing the gradient wrt the input, it seems natural to try to estimate it as in ProjForward using vjp with  random directions. Have you thought that or tried it?
>
> Thank you for the great suggestion! We indeed tried that option but without much success. We believe that if we estimate it with multiple directions then when we multiply with inverse matrix the error would quickly grow. We think that more constraints are needed in order to solve this problem. In general, we think that this idea might be another excellent point to prefer moonwalk instead of backprop/reversible.
>
> 2. Although it seems nice to extend the applicability of Moonwalk to a larger class of functions, I find it hard to get intuition of what this changes in the context of deep learning models. Do you have examples of layers which would be submersive but non invertible?
>
> Thank you for your comment, we did include the examples. Algorithm 2 showcases a network with linear layers. Basically, any network with linear layers or 1d convolutions that have output is smaller than its input would be submersive, but not invertible. Another point is that we need to add constraints like making linear layers upper triangular with ones on the main diagonal to make it stable and invertible with Gaussian elimination.

---

> ### Comment · Reviewer_Gc2t · 2024-12-02
>
> I thank the authors for answering my questions.
>
> While overall I like the proposed method and find it elegant, I do not find the experimental section convincing enough.
>
> I am keeping my score as it is.

---

### Official Review · Reviewer_i14z · 2024-11-03

**Soundness:** 2
**Presentation:** 3
**Contribution:** 2
**Rating:** 5
**Confidence:** 3

**Summary:**

The paper explores alternative gradient computation strategies in the context of invertible neural networks, leveraging the properties of surjective differentials to reformulate the chain-rule recursion for parameter derivatives. A two-stage semi-forward-mode gradient computation algorithm, named Moonwalk, is proposed to reduce memory overhead in gradient computation. This approach requires multiple forward passes to determine the input gradient and perform forward differentiation recursion. The authors provide a complexity analysis of different gradient computation methods to highlight the potential of Moonwalk. Experiments on classical image classification tasks are conducted to evaluate the algorithm's performance comprehensively.

**Strengths:**

This paper presents a simple yet practical approach to managing the cost of forward differentiation, highlighting a direction distinct from subspace projection and offering the potential for further exploration. The authors provide a clear and comprehensible description of the methodology and present detailed theoretical properties to demonstrate its advantages.

**Weaknesses:**

My main concern is a potential paradox: if activation gradient computation in Backprop is much more expensive than parameter gradient computation, the Mix algorithm’s first step yields minimal savings; otherwise, activation storage costs can become negligible. This could place the Mix variant, which is essential for showcasing Moonwalk's advantages, in an awkward position.

The paper attempts to demonstrate the advantages of the proposed algorithm through experiments from multiple perspectives. However, certain aspects of the experimental setup and presentation are suboptimal, which affects the demonstration of the algorithm's effectiveness. Please refer to the questions section for further details.

**Questions:**

- In Section 4.2, the authors mention that Backprop typically retains some information that could be discarded. It is not due to Backprop itself but rather to optimize computation pipeline utilization (see, e.g., [1]). I am curious whether, when the authors consider pipeline scheduling efficiency across multiple data batches for Moonwalk, similar retention of additional information might occur, as observed in Backprop.

- The tests on time and memory overhead require more careful execution. In Figures 4 and 7, when each block contains three layers, the time consumption deviates from a monotonic trend, which seems unexpected and lacks explanation.

- Figures 2-7 contain instances where figure captions do not match the content, and references in the text are incorrect or entirely missing, significantly hindering the readability of Section 6.

- The experiments involve up to five baselines, so why do most results include only two or three of them? Except for the vanilla forward algorithm, which may be prohibitively costly, the remaining methods should be testable within a reasonable timeframe.

- The learning curves in Figures 3 and 7 lack key hyperparameter descriptions, raising concerns about whether the results are consistent under alternative experimental settings.

>[1] Narayanan, D., Harlap, A., Phanishayee, A., Seshadri, V., Devanur, N. R., Ganger, G. R., ... & Zaharia, M. (2019, October). PipeDream: Generalized pipeline parallelism for DNN training. In Proceedings of the 27th ACM Symposium on Operating Systems Principles (pp. 1-15).

---

> ### Author Response · Authors · 2024-11-26
>
> 1. "My main concern is a potential paradox: if activation gradient computation in Backprop is much more expensive than parameter gradient computation, the Mix algorithm’s first step yields minimal savings; otherwise, activation storage costs can become negligible. This could place the Mix variant, which is essential for showcasing Moonwalk's advantages, in an awkward position."
>
> Thank you for your comment. We would like to highlight using an example of an algorithm 2 vs 3 for submersive networks, which we included in the appendix. For further clarification, we also included an example of training such a network with linear layers or with 1d convs. In case when activation gradient computation is very expensive, we show that we need only to store signs, whereas for backprop we have to store the entire variable. This basically means that we are reducing memory footprint from fp16/32 to binary i.e 16x/32x more memory efficient. We would like to highlight, that it would be significantly more efficient for convolutions. We also include a code fo efficient matrix inverse (Avoiding using SVD) with Gaussian - Elimination.
>
> 2. "The paper attempts to demonstrate the advantages of the proposed algorithm through experiments from multiple perspectives. However, certain aspects of the experimental setup and presentation are suboptimal, which affects the demonstration of the algorithm's effectiveness. Please refer to the questions section for further details."
>
> We do agree that some of the experiments were not optimal. We would like to also clarify that the main point is not to show numerical stability, but rather to show that we can train submersive networks, whereas reversible backprop can not do that.
>
> Questions:
>
> 1. In Section 4.2, the authors mention that Backprop typically retains some information that could be discarded. It is not due to Backprop itself but rather to optimize computation pipeline utilization (see, e.g., [1]). I am curious whether, when the authors consider pipeline scheduling efficiency across multiple data batches for Moonwalk, similar retention of additional information might occur, as observed in Backprop.
>
> Thank you for the reference. We do agree that by trading-off memory/computation we can come up with different graphs based on user needs. We would also like to highlight that based on algorithm 3 (appendix) we would only need to store signs instead of full variables for computing gradients wrt input. We would also like to highlight the potential benefit of Moonwalk in the context of multi-gpu training. If we split the model across multiple GPUs we would experience a bottleneck, but in the case of moonwalk the first phase would be faster, and the potential bottleneck would be smaller.
>
> 2. The tests on time and memory overhead require more careful execution. In Figures 4 and 7, when each block contains three layers, the time consumption deviates from a monotonic trend, which seems unexpected and lacks explanation.
>
> We do agree with the reviewer. We are gonna address this point in the draft. The main problem is that JAX on CUDA does not guarantee optimal memory allocation, some times it can construct graphs with less memory, but with more computation.
>
> 3. Figures 2-7 contain instances where figure captions do not match the content, and references in the text are incorrect or entirely missing, significantly hindering the readability of Section 6
>
> Thank you for the comment, we clarified the figure notation in the updated version.
>
> 4. The experiments involve up to five baselines, so why do most results include only two or three of them? Except for the vanilla forward algorithm, which may be prohibitively costly, the remaining methods should be testable within a reasonable timeframe.
>
> We did not include projForward mostly because of its inaccurate gradient estimation. We would like to highlight that this method does not produce accurate gradients, and in all our experiments it failed to train an end-to-end network.
>
> 5. The learning curves in Figures 3 and 7 lack key hyperparameter descriptions, raising concerns about whether the results are consistent under alternative experimental settings.
>
> Thank you for your point. We will add hyperparameters to the appendix.

---

> > ### Comment · Reviewer_i14z · 2024-12-02
> > **Response to Rebuttal**
> >
> > I thank the authors for their response to my concerns. While revisions have been made, the overall presentation, including the new sections, could be clearer. There are still several typos, and some content remains undefined, misreferenced, or insufficiently explained within the context. My concerns regarding the numerical results have not yet been adequately addressed. For instance, the effect of irrelevant factors could be alleviated through careful configuration. Therefore, my original assessment remains unchanged.

---

### Official Review · Reviewer_zPRR · 2024-11-06

**Soundness:** 2
**Presentation:** 4
**Contribution:** 2
**Rating:** 5
**Confidence:** 5

**Summary:**

The presented paper proposes a new algorithm for estimating gradients of reversible neural networks, able to decrease the memory footprint compared to standard backpropagation. The authors further claim a superior numerical stability than reversible backpropagation, hence showing a potential advantage of their method for reversible architectures. The paper is generally well-written, easy to understand, with a careful time and memory complexity analysis with hypothesis clearly stated, from both a theoretical and practical point of view.

**Strengths:**

1. The paper is very well-structured, easy to read and understand. Apart from very few notations that could be improved, this is a very appreciated feature of the paper.
2. The authors carefully compare their proposed method with relevant alternatives aiming at trading time and memory complexities for neural network training.
3. Experimental setup are clearly detailed and figures are very easy to read, which is highly appreciated.

**Weaknesses:**

1. Section 3.1 gives explicit notations for the different quantities involved along with their dimensions. However, the dimension of parameters $\theta$ or $\theta_i$ is not mentioned. The reviewer assumes that for all $1 \leqslant i \leqslant L$, the parameters $\theta_i$ have dimensionality $d$. This might be important when discussing complexity issues down the line. Similarly at line 153-154, the reviewer assumes that “the suffixes are of dimension $n_i \times d$. Note that in section 5, authors assumes that all these quantities are the same for all $i$ for simplification. Either this simplification can be done in section 3.1, either all quantities should depend on $i$ in section 3.1 and then simplified in section 5.

2. In section 4.2, authors argue that using backprop to compute the gradient $h_0$ only can lead to substantial memory savings. This claim stems from two separate arguments:
- The first is that if we only want to compute $h_0$ , we do not need to store any activation computed from parameters $\theta_i$ independent from $x_i$. While this is true, I would like the authors to give some examples of architectures where this argument would be relevant.
- The second argument is that parameter gradients can take up a substantial portion of memory during backpropagation. This argument needs to take into account concrete details about the computational task at hand. While it is true that gradients can take up substantial memory space, one can resort to “fused optimizer” as a way to decrease the peak memory footprint of each iteration, which consists in applying parameter update for layer $i$ before continuing backpropagation on layer $i-1$. This would however only be possible if no gradient accumulation is needed, otherwise, gradients would need to be stored between different micro-batches. However, if gradient accumulation is needed, then mixed-mode random walk would also need to preserve gradients between micro-batches, rendering the argument ineffective. The only remaining argument would be to show that the memory reduction of mixed-mode moonwalk is such that we can increase the batch size to a point where no gradient accumulation would be needed, while it would still be necessary in standard backpropagation or RevBackprop. Overall, this makes the memory reduction argument of mixed-mode moonwalk fairly weak.

3. In the paragraph Memory complexity with checkpointing in section 5, the notation $n$ representing a bound on each layer’s size should be named differently, as it refers to the dimension of $x_0$ in section 3.1. Furthermore, $c$ must be lower than $\frac{L}{c}$, and it is not clear to the reviewer how the best tradeoff $c$ stated at line 309 can be guaranteed to be lower than $L$, unless $M_x + M_{\theta} \leqslant n$ or at least $M_x + M_{\theta} = \mathcal{O}(n)$.

4. The authors show a superior numerical stability with the use of a TanH function. First, the reviewer does not understand why the authors resort to this activation function w.r.t model performance. In absence of a better justification for the use of that function, it looks like this activation has been chosen to favor their method against others. Please provide an argument for the relevance of this activation function. The first reviewer’s guess is that authors wants to use a reversible activation function, but they could as well resort to softplus or leakyRelu as an alternative to ReLU. Furthermore, the reviewer is not aware of reversible residual networks whose activation function is applied on both streams. Instead, in most reversible architectures, the non-linearity are all embedded into the function $\mathcal{F}$. As far as the reviewer understands, RevBackprop has a lower memory footprint than Mixed-mode Moonwalk, which means that the numerical stability argument would be the only remaining argument to justify the use of the proposed method. Thus, it would be nice to elaborate to what extent this numerical stability advantage would be crucial within modern architectures.

5. Line 417: why do the authors pad the input with zeros in the channel dimension ? This increases the input dimension by ~2.6 or 6. The reviewer would like some explanation. Is it solely to experiment with different kinds of input size "synthetically" ?

6. Albeit authors do investigate time and memory tradeoff, their focus on simple datasets and architectures does make it easier to analyze their method in detail on the presented examples, but it would be nice to summarize the practical potential of the proposed method. For example, it would be convenient to summarize the scaling potential offered by the proposed methods against RevBackprop on use cases where memory footprint is the crucial limiting factor. While this might not represent substantial text, it might be highly valuable to the reader.

**Questions:**

Could the author address the weaknesses in general ?
The reviewer would appreciate a detailed explanation on weakness 2 and 4 specifically.

---

> ### Author Response · Authors · 2024-11-26
>
> We sincerely thank the reviewer for their constructive feedback and valuable suggestions, which have greatly helped us refine our manuscript and address potential ambiguities.
>
> Firstly, we would like to clarify that the primary focus of our work is to introduce a novel method for computing gradients using forward mode, specifically designed for submersive networks. Please, see our message to all reviewers. To underscore the significance of our approach, we have added an illustrative example in the revised manuscript of a submersive network that cannot be trained using reversible backpropagation but can be successfully trained with Moonwalk.
>
> To clarify further, any linear layer with output dimensionality  \(k \leq n\)can be considered a submersive layer. Notably, reversible networks are incapable of effectively training such networks. Our primary contribution lies in introducing a novel method for efficiently training submersive networks, overcoming the limitations faced by reversible networks.
>
> 1. “The dimension of parameters are not mentioned”:
>
> Thank you for your comment. For submersive networks, in general, the input size is \(n\), but for all subsequent layers, the output size of each layer can be \(k \leq n\). For simplicity, we assumed that all layers have a fixed parameter size of \(d\) and input-output size of \(n\). We will clarify this assumption in Section 3.1.
>
> 2. “An example of architecture”.
>
> Here, we provide a simple yet widely-used architecture as an example. Technically, any sequence of linear layers with decreasing dimensionality can serve as a representative case. To illustrate this, we have included a code snippet that highlights the architecture.
> The key point is that in standard backpropagation, the entire activation (\(z_2\)) must be stored. In contrast, with our proposed Moonwalk method, when computing \(h_0\), we only need to store the signs of this activation. If a function like leaky-ReLU is used, this allows us to reduce storage from \(fp16\) to binary for every number in \(z_2\), leading to a potential 16x memory savings during this phase.
>
> 2. "fused optimizer"
>
> Could you please clarify which paper you are referring to? If you are discussing in-place updates, there are several potential issues with such networks that can negatively impact convergence performance. Specifically, if the gradient for \(L_{n_1}\) depends on \(W_n\), updating \(W_n\) prematurely can result in incorrect gradient estimation, which could be a case if we talk about the same weights in the block with residual connections, for example.
>
> 3. Could you please clarify this point “c should be lower L/c”?
>
>
> 4. "The authors show a superior numerical stability with the use of a TanH function"
>
> The tanh activation function is just one of many examples that illustrate the limitations of reversible models. We are not the first to point out instability in reversible models under certain conditions, as discussed in [https://arxiv.org/pdf/2306.00477]. However, we acknowledge that using a single activation function is insufficient to fully demonstrate the broader stability issues.  We would also like to point out that, it is not the main point in the comparison, since we can operate on a wider class of networks. Please, see our message to all reviewers.
>
> 5. "Line 417: why do the authors pad the input with zeros in the channel dimension"
>
> The primary goal is to increase the effective dimensionality of the model. A key limitation of bijective invertible networks is that they require the same input and output size for every layer. Here we use reversible architecture as shown in RevNet paper. For example, in an MNIST classification problem with an input size of 64, the network would be constrained by having all layers fixed at size 64, creating a bottleneck.  To address this, one approach is to pad the input with zeros, effectively increasing the network’s capacity. Another way to conceptualize this is by projecting the input into a higher-dimensional space. In general, our method is more flexible, since we can have contracting layers.
>
> 6. "Albeit authors do investigate time and memory tradeoff"
>
> We want to emphasize that our approach enables the training of submersive networks, which is not possible with RevBackprop. To illustrate this, we have included an example along with a code snippet. Additionally, our method outperforms both standard backpropagation and checkpointing when applied to submersive networks—a broad and versatile class of network architectures.

---

> ### Comment · Reviewer_zPRR · 2024-12-02
>
> I thank the authors for their detailed response. I would like to make a few more comments:
>
> 1. **About parameter dimensionality.**
>
> I did not keep the original version of the article so I am not sure what have been updated in section 3.1.
>
> 2.  **About memory savings.**
>
> I thank the authors for their simple yet relevant example of an architecture where mixed-mode moonwalk offers substantial memory savings compared to standard backpropagation. Still, memory-wise, the advantage is not clear to the reviewer when considering reversible architectures since reversible backpropagation does not require to store intermediate activations. The memory advantage is however straightforward for submersive architectures.
>
> The reviewer however has some concerns about submersive architectures. While it might be feasible to prove that random $k \times n$ matrices have a high probability of being subjective, some computational tasks might collapse the rank of the matrix below $k$. In practice, matrices are rarely low rank, but often ill-conditioned with many singular values close to zero, which might affect numerical stability when computing a right-inverse. While a thorough ablation of each of the problem associated with numerical stability would be expensive, it would still be nice to spend some time either in the main body, either in the appendix, to emphasize how much wider is the class of submersive networks compared to the class of reversible networks, especially with respect to the current literature. For example, reversible architectures often require to maintain a constant dimensionality; this makes it non-trivial to adapt ResNets to fully-reversible architectures due do downsampling operations.
>
> 2. **About fused optimizer.**
>
> The reviewer were not aware of papers studying fused optimizer specifically, but this reference seems to be the one I would be looking for. Fused optimizer were proposed to optimize memory consumption during training in the Apex library. As reviewers correctly points out, updating the weights too soon might lead to incorrect computations for subsequent gradients, but it is often possible to apply this update way before the end of the full end-to-end backpropagation. Nevertheless, the reviewer does not believe that this is a major concern for mixed-mode Moonwalk specifically and does.
>
> 3. **"$c$ must be lower than $\frac{L}{c}$"**
>
> I am sorry, this was a typo. I meant "$c$ must be lower than $\frac{L}{n}$".
>
> 4. **About numerical stability.**
>
> The reviewer slightly disagree with the author with respect to the importance of their numerical stability claim. The reviewer acknowledge that their method is applicable to a wider class of model, but they did not focus their experiments on submersive architectures that are not reversible. Therefore, the reviewer took the claim of numerical stability seriously. The reviewer think that if numerical stability is such a serious issue, it should be better explained and supported with references or experiments in the paper.
>
>
> **My review update**: Overall, the authors answer one of my main concern, which was to emphasize the memory reduction offered by mixed-mode moonwalk compared to backpropagation. The fact that mixed-mode moonwalk is applicable to submersive networks that are not reversible should be emphasized more in the experiments; for example, there is no mention of the cost of the inversion and how it affects training. The reviewer do not see many advantage over reversible backpropagation for reversible networks, apart from the improved numerical stability claim that the reviewer do not find convincing *yet*.
>
> I am increasing my score from 3 to 5 given that the memory constraint is the main limiting factor. Should the author emphasize better the practicality of submersive architecture or the relevance of improved numerical stability, I'd be willing to improve my score further.

---

### Author Response · Authors · 2024-11-26
**Comments to all reviewers**

We thank all reviewers for their insightful comments and the opportunity to improve our work.

First, we would like to highlight the primary benefit of our method: its applicability to submersive networks. We acknowledge that the previous version of the paper lacked concrete examples of networks that are submersive but not invertible, where reversible backpropagation is not applicable. To address this, we have included illustrative examples in the revised manuscript to clarify these distinctions.

Second, we emphasize the computational advantages of our method. Algorithms 2 and 3 in the manuscript outline the key differences in the computational graph. Unlike backpropagation, which requires storing all intermediate variables, our method (Moonwalk) reduces storage requirements. Specifically, for gradient computation, Moonwalk only requires the storage of signs for activation functions, providing a clear advantage in terms of memory efficiency.

In the updated version of the manuscript, we added:
Architecture for showcasing a submersive non-invertible network with algorithms to train them using Moonwalk. Algorithm 3,4,5

The section about using submersive networks for training. Sections 6.6 and 6.7.

Updated captions on figures 2-4.

We added examples of SVD and Gaussian Elimination for effective matrix inversion.

We updated references to acknowledge new work in the field of checkpointing.

---

### Meta-Review · Area_Chair_1xc9 · 2024-12-20

**Metareview:**

The paper develops Moonwalk, a forward-mode differentiation method for *submersive* networks, i.e. networks with surjective Jacobians. In particular, all invertible networks are submersive. The method requires computing the gradient with respect to the input first, but then the rest of the computation is efficient in forward mode. In theory, forward-mode differentiation does not require storing the network activations, and can substantially reduce the memory footprint of gradient computation compared to standard backpropagation. Apart from computing the gradient with respect to the input, Moonwalk also matches backpropagation in terms of time complexity. In the experiments, the authors compare the proposed method to several baselines (Backprop, RevBackprop, ProjForward) on both runtime and memory, with the mixed method (use Backprop for input gradient and Forward mode for the rest) showing good results.

Strengths:
- Novel forward propagation method with interesting properties.
- The authors identify a broader class of applicability for their method compared to RevBackprop: submersive networks.
- In the experiments, the Moonwalk shows good performance on both runtime and memory compared to baselines.

Weaknesses:
- The fully-forward variation of MoonWalk is still impractical (five days to train on CIFAR-10)
- The mixed version involves Backprop to compute gradient with respect to the input. The authors show that in practice the backprop for just the input gradient can be cheaper in terms of memory compared to full backprop.
- Compared to RevBackprop, the main advantage is the applicability to submersive but non-invertible networks. However, the experiments focus on invertible RevNet models.
  + For the comparison to RevBackprop, the authors show a numerical instability on the RevBackprop; however, the setup involves an unusual activation function that may be specifically chosen to increase RevBackprop instability
- The method is not generally applicable, it requires the network architecture to be submersive.

Decision recommendation: The paper makes an interesting methodological contribution. Mixed Moonwalk is potentially an interesting method for submersive but non-invertible models. However, currently the experiments fail to highlight the performance on that model class, as they focus on invertivle models. For invertible models, it is unclear if the method has advantages over RevBackprop. I believe the authors should emphasize results on submersive but non-invertible models. In the current form, I am leaning towards rejecting the paper.

**Additional Comments On Reviewer Discussion:**

The reviews were mixed, with three out of four suggesting reject: 6, 3, 5, 5. The reviewers highlighted that the method does not have obvious advantages over RevBackprop for invertible models, and that it is not clear how often submersive but non-invertible architectures are used in practice. Authors provided detailed responses, but three of the reviewers remained unconvinced.

---

### Decision · Program_Chairs · 2025-01-22

Reject